# Affordable Production of Antioxidant Aqueous Solutions by Hydrodynamic Cavitation Processing of Silver Fir (*Abies alba* Mill.) Needles

**DOI:** 10.3390/foods8020065

**Published:** 2019-02-12

**Authors:** Lorenzo Albanese, Alessandra Bonetti, Luigi Paolo D’Acqui, Francesco Meneguzzo, Federica Zabini

**Affiliations:** 1Institute of Biometeorology, National Research Council, 10 Via Madonna del Piano, I-50019 Sesto Fiorentino (FI), Italy; l.albanese@ibimet.cnr.it (L.A.); f.zabini@ibimet.cnr.it (F.Z.); 2Institute for Research on Terrestrial Ecosystems, National Research Council, 10 Via Madonna del Piano, I-50019 Sesto Fiorentino (FI), Italy; alessandra.bonetti@cnr.it (A.B.); luigipaolo.dacqui@cnr.it (L.P.D.)

**Keywords:** *Abies alba* Mill., antioxidant activity, coniferous trees, essential oils, flavonoids, food preservation, green extraction, hydrodynamic cavitation, nutraceutics, polyphenols

## Abstract

Extracts from parts of coniferous trees have received increased interest due to their valuable bioactive compounds and properties, useful for plenty of experimental and consolidated applications, in fields comprising nutraceutics, cosmetics, pharmacology, food preservation, and stimulation of plant growth. However, the variability of the bioactive properties, the complexity of the extraction methods, and the use of potentially harmful synthetic chemicals, still represent an obstacle to the spreading of such valuable natural compounds. Hydrodynamic cavitation is emerging as a promising innovative technique for the extraction of precious food components and by-products from waste raw material of the agro-food production chain, which can improve processing efficiency, reduce resource consumption, and produce healthy, high-quality products. In this study, a process based on controlled hydrodynamic cavitation was applied for the first time to the production of aqueous solutions of silver fir (*Abies alba* Mill.) needles with enhanced antioxidant activity. The observed levels of the in vitro antioxidant activity, comparable or higher than those found for reference substances, pure extracts, and other water extracts and beverages, highlight the very good potential of the hydrodynamic cavitation (HC) process for the creation of solvent-free, aqueous solutions endowed with bioactive compounds extracted from silver fir needles.

## 1. Introduction

Fir needles from different plant varieties share a long history of beneficial use for human health and other applications, deriving from the anti-inflammatory and antiseptic properties, later attributed to the respective essential oils (EOs), polyphenols and flavonoids, and linked to their antioxidant activity. Reports exist, dating back to the 16th century, about the use of fir parts for prophylaxis and treatment of various diseases [1,2], as well as for recreational purposes, in particular in spruce beer [3,4]. Indeed, spruce beer turned out to be an effective preservative against, and remedy for, scurvy [5], as well as for rheumatic joints, colds and venereal disease [6], with its use reported in many areas of the world [4,5,7,8,9,10].

Nowadays, the genus *Abies* of the *Pinaceae* family consists of approximately 50 species, which are largely distributed in temperate and boreal regions of North and Central America, Europe, Asia, and North Africa, eight of which are endemic to the Mediterranean region [11]. The silver fir (*Abies alba* Mill.) species, which is of special interest to this study, is widespread in central Europe, as well as in mountainous regions such as Pyrenees, Carpathians, Balkans, Alps, and Italian Apennines at relatively high altitudes [12].

As of the late 2000s, plenty of studies had been carried out on the chemical composition of firs (*Abies* genus), leading to the identification of several secondary metabolites such as terpenoids, flavonoids, phenols, lignans, steroids, totalling 277 compounds from 19 plants (out of about 50 known worldwide) of *Abies* species, among which *Abies alba* Mill. Further information about the chemical composition of silver firs, and the related variability, is available in Appendix A.

The study of chemical composition, biological and pharmacological activities of the extracts isolated from needles, bark, and wood from different coniferous tree species, experienced renewed interest in recent decades, aimed at applications in the fields of nutrition, health and medicine [13]. Extracts from most of the fir species exhibited a wide range of remarkable biological activities, such as antiproliferative, antibacterial, anti-inflammatory, cardiovascular, and central nervous system activities, at least some of them likely mediated by the respective antioxidant activity [14].

Most of the studies focused on essential oils extracted from parts of fir plants by means of different techniques, such as steam distillation (the dominant technique), maceration, solvent extraction, cold pressing, supercritical fluids, rectification, enfleurage (use of odourless animal or vegetable fats), solid phase extraction, etc. [15]. On the other hand, the extraction of phenolic compounds made use of solvents such as methanol, ethanol, acetone, diethyl ether and ethyl acetate, which are potentially hazardous for the human health, often mixed with different proportions of water [16]. Beyond the inherent complexity, and the use (and subsequent removal) of potentially harmful chemicals, such selective extraction techniques need high processing temperatures, and long process times. Few phytochemical and biological studies focused also on aqueous extracts, showing interesting findings, which are especially relevant to this study. Further information about recent studies is available in Appendix B.

In most of the studies shortly discussed in Appendix B, the aqueous solutions including fir parts were manufactured by means of relatively complex processes. They included pulverization of the raw material, extraction in water (also tap water) by means of different techniques, concentration by drying (e.g., vacuum drying), sometimes filtering and/or lyophilisation, and finally, dispersion of the resulting material in tap or distilled water.

In this study, for the first time, a solvent-free, fast, unselective, scalable extraction process was tested for its potential to produce aqueous solutions based on silver fir (*Abies alba* Mill.) needles and boost the respective in vitro antioxidant activity. This process was based on controlled hydrodynamic cavitation (HC), an emerging technique with a short but successful history in the extraction of valuable biocompounds [17,18,19]. Beyond the resulting biologically relevant properties, the manufacturing process of the aqueous solutions is very important, in particular the respective scalability and cost, conditioning the overall feasibility, as well as the affordability of the final product.

The HC equipment did not include any proprietary component, whole fir needles were processed in tap water at low to moderate temperatures, without previous grinding or pulverization, and centrifuge separation was applied to isolate the liquid phase from the residual insoluble solids. Beyond the scientific interest, the proposed method could lead to practical developments, related to products of interest to the nutraceutical and pharmaceutical industries, as well as to applications in the fields of food conservation, plant growth stimulation, and forestry.

## 2. Materials and Methods

### 2.1. HC Device and Method

Figure 1 shows the experimental device implementing the HC-based process, including a closed hydraulic loop (total volume capacity around 230 L) and a centrifugal pump (7.5 kW nominal mechanical power, rotation speed 2900 rpm). Such a device was used to produce beer wort in past studies, to which reference is made for any detail, including all components and measurement devices [19,20,21,22], as well as the geometry of the Venturi-shaped cavitation reactor [23].

The cavitation process is associated with the heating of the liquid-solid mixture due to the thermal conversion of the pump’s mechanical energy [19]. The tests were carried out under atmospheric pressure.

Theoretical background on the hydrodynamic cavitation processes was comprehensively exposed in many articles and reviews [18,19,20], with mechanical methods recognized as comparatively more efficient, robust and scalable [24]. As well, theoretical and experimental evidence has grown about the unique physical (mechanical and thermal) phenomena occurring at the scale of the collapsing cavitation bubbles [25,26], and the chemical phenomena, such as water splitting and generation of powerful oxidants (e.g., OH⋅hydroxyl radicals) [25,27], the latter however quite limited in the absence of specific oxidizing additives [28,29].

The main metric of HC processes, i.e., the cavitation number (*σ*), was defined long ago [30]. It is a dimensionless parameter, derived from Bernoulli’s equation, and representing the ratio between the pressure drop needed to achieve vaporization, and the specific kinetic energy at the cavitation inception section, as per Equation (1):σ = (P_0_ − P_v_)/(0.5·ρ·u^2^)(1)
where P_0_ (Nm^−2^) is the average recovered pressure downstream of a cavitation reactor, such as a Venturi tube or an orifice plate, where cavitation bubbles collapse. Since the fluid was not pressurized, P_0_ was assumed equal to the atmospheric pressure. P_v_ (Nm^−2^) is the liquid vapor pressure (a function of the average temperature for any given liquid). ρ (kgm^−3^) is the liquid density, and u (ms^−1^) is the flow velocity through the nozzle of the cavitation reactor, the latter also depending on the pump’s inlet pressure. In this study, the values of the cavitation number during the processes were computed according to the available data, such as temperature and pump discharge; the latter were retrieved based on the consumed power, as explained in a previous study [20].

The cavitation number has recently been placed under serious scrutiny over its representativeness and, in general, the reproducibility of HC processes [31]. However, under certain conditions, easily achievable with Venturi-shaped reactors, developed cavitation, with strong and frequent collapses of the cavitation bubbles, arises whenever the cavitation number falls within the range 0.1 to 1, and even at greater values in the presence of solid particles or dissolved gases [32,33].

### 2.2. Silver Fir Needles Samples and Tests

Needles-carrying young twigs from lower branches of three adjacent silver firs, in practically identical amount from each tree (±5% in mass), were manually collected in mid-October 2018. The silver fir trees belonged to a mixed beech tree—fir forest in northern Tuscany, Italy, near the village of Careggine (44°06′15″ N, 10°20′ E), at an altitude of about 1000 m a.s.l. on the north-eastern slopes of the Apuan Alps mountain range. The average annual climate was characterized by over 2000 mm rainfall [34], and temperature around 10 °C [35]. Local fir species included silver fir and Douglas fir (*Pseudotsuga menziesii*), however only twigs from silver firs were harvested, and the respective needles used for the tests. Figure 2 shows a view of a portion of the fir forest (a), and a detailed view of silver fir twigs and needles (b).

Needles were manually removed from the twigs, until a batch of 5 kg was formed. Then, the same batch was gently mixed in a vessel by means of a magnetic stirrer at the minimum speed of 50 rpm, in order to preserve the needles integrity, while ensuring the batch homogeneity, and the representativeness of any extracted sample.

The first needles sample was extracted the day after the harvesting, with a water content of 30%; since its mass was 0.756 kg, the dry weight was 0.529 kg. The second needles sample was extracted from the same homogenous batch, 40 days after the harvesting, during which the batch was kept at a constant temperature of 30 ± 1 °C, resulting in a water content of 2%. The mass of the second needles sample was 0.540 kg, resulting in the same dry weight as the first sample, i.e., 0.529 kg.

Two tests were performed, hereinafter referred to as Silver Fir Needles—Test 1 (SFN_T1) and Silver Fir Needles—Test 2 (SFN_T2), both using 120 L of tap water as the only solvent, as well as carried out under atmospheric pressure. The test SFN_T1 used the first needles sample, while the test SFN_T2 used the second sample, therefore the silver fir needles were mixed with tap water in concentrations of 0.44% (*w*/*w* dry basis). Such relatively low concentration was dictated by the need to prevent clogging of the closed impeller pump; as explained in a previous study dealing with HC-based biochar modification [36], such concentration could easily be increased with the use of an open impeller pump.

The whole needles were mixed with water in the HC device from the beginning of each test, with SFN_T1 undergoing unconditioned heating from 27 °C to 67.5 °C during 90 min, resulting from the balance between the mechanical energy supplied by the pump impeller and the heat loss through the uninsulated walls of the hydraulic circuit. In SFN_T2, the heating was unconditioned from 31.5 °C up to the temperature of 43 °C, during 30 min, then a further isothermal step was carried out at 43 °C during 30 min, with cooling water flowing in a jacket surrounding the HC device to remove the excess heat.

Each sample, collected during the tests, was immediately centrifuged (3800 rpm, 10 min), and the supernatant was extracted for the measurements. The test samples were kept refrigerated in the dark at 4 °C, and measured the day after each test, as well as later for stability assessment.

### 2.3. Analytical Procedures

#### 2.3.1. Total Phenolic Content

The total phenolic content (TPC) was determined with the standard Folin-Ciocalteau assay [37]. A sample volume of 500 mL was added to 1 mL of distilled water in a 25 mL test tube. A volume of 2.5 mL of Folin-Ciocalteau reagent was added, and after 5 min, a volume of 5 mL of Na_2_CO_3_ anhydrous solution was added, along with distilled water. Samples were read after 2 h.

TPC was expressed as mg of Gallic acid equivalent per mL of aqueous solution (mgGAE/mL). A reference curve was prepared with Gallic acid concentration ranging from 0.1 mg/mL to 0.5 mg/mL, and the solutions read at the wavelengths of 715, 730, and 760 nm. Three curves were built, and the best fitting one, with R^2^ = 0.9995, was the one corresponding to 730 nm, which was therefore assumed as the standard curve.

The absorbance of the colored reaction product was read at the 730 nm standard curve using a Varian UV-Visible spectrophotometer Cary 50 Scan. Each analysis was performed in triplicate.

#### 2.3.2. Total Flavonoid Content

The total flavonoids content (TFC) was measured according to a standard procedure [38]. A quantity of 4 mL of distilled water, and 300 μL of NaNO_2_ (5%), were added to 1 mL of water extracts of fir needles, and the samples were allowed to stand for 5 min. Subsequently, 300 μL of AlCl_3_ (10%) were added, and samples were left to stand for 6 min. Then, 2 mL of NaOH 1 M were added to stop the reaction, and samples were brought to a final volume of 10 mL with distilled water. Samples were read after 15 min.

TFC was expressed as mg of catechin equivalent (CE) per mL of aqueous solution (mgCE/mL), using an equation obtained by standard calibration graph with (+)-catechin. A reference curve was prepared with catechin concentration ranging from 0.02 to 0.1 mg/mL, and solutions were read at the wavelengths of 500, 510, and 530 nm. Three curves were built, and the best one with R^2^ = 0.9998 was the one corresponding to 510 nm, which was therefore assumed as the standard curve.

Samples were read at the 510 nm standard curve using a Varian UV-Visible spectrophotometer Cary 50 Scan. Each analysis was performed in triplicate.

#### 2.3.3. DPPH Radical Scavenging Assay

The 2.20-diphenyl-1-picrylhydrazyl (DPPH) assay is commonly applied for the estimation of the antioxidant activity of plant extracts [39], and is related to the respective activity against lipid oxidation, which in turn affects food shelf life and human health [12,40]. The DPPH radical-scavenging activity was determined using a standard method [41].

A volume of 1 mL of DPPH solution was added to 1 mL of serially diluted samples. The obtained solution was read at a wavelength of 517 nm, immediately after the addition of DPPH solution, and after keeping samples for 20 min in the dark, in order to measure the decreased absorbance of extract (AE).

The radical scavenging activity was calculated by the percentage of DPPH that was scavenged, using Equation (2):% Reduction = [(AB − AE)/AB]·100(2)
where AB is the absorbance of the blank sample, while AE is the absorbance of the water extracts. The Effective Concentration values (IC50), defined as the amount of antioxidant required to scavenge DPPH radicals by 50%, were calculated from the results, and expressed as μg per mL of aqueous solution (µg/mL). Each analysis was performed in triplicate.

#### 2.3.4. Oxygen Radical Absorbance Capacity (ORAC) Assay

The oxygen radical absorbance capacity (ORAC) assay measures the scavenging of peroxyl as well as of hydroxyl radicals, and has been successfully applied to the assessment of antioxidant species in human plasma [42]. The ORAC measurement method was adapted from a previous work [43]. The instrument was a fluorescence spectrophotometer (Varian Cary Eclipse, Palo Alto, CA, USA). The sample (150 μL) was added to a free-radical generator (AAPH, 2, 2′-azobis (2-aminopropane) dihydrochloride) (75 μL), and the inhibition of the free radicals was measured. Fluorescein (2.74 mL) was used as a target for free-radical attack. The exciting wavelength was 490 nm and the emission wavelength was 512 nm. Total antioxidant capacity, as measured by ORAC, was obtained using Equation (3):ORAC = 20·k·(S_s_ − S_b_)/(S_Trolox_−S_b_)(3)
where k is the dilution factor, S_s_ is the area under curve area of the sample, S_b_ is the area of the blank under the curve, and S_Trolox_ is the area of the standard (Trolox, 1 μM, 150 μL) under the curve. ORAC values were expressed as μM Trolox equivalents per Liter of aqueous solution (µMTE/L), using the standard curve established previously. Each analysis was performed in triplicate.

## 3. Results

### 3.1. Main Operational Parameters

The cavitation number, computed according to Equation (1) and the method mentioned in Section 2.1, was in the range of 0.3 to 1.1 in both tests SFN_T1 and SFN_T2, thus falling in the range of developed cavitation, as defined in Section 2.1. Based on the estimated average flow (330 L/min), and the water volume (120 L), the average frequency of passages through the cavitation reactor was about 2.75 per min.

The average power absorbed during SFN_T1 was around 5400 W, with a total electricity consumption, during 90 min of process time, of 7.9 kWh. In test SFN_T2, the two quantities reduced to 4900 W, and 4.56 kWh (in 60 min of process time), respectively.

### 3.2. Total Phenolic and Flavonoids Content

Figure 3a shows the joint evolution of temperature and TPC, for tests SFN_T1 and SFN_T2. The same holds for Figure 3b, except that the evolution of TFC is represented.

In SFN_T2, the evolution of the total phenolic content exhibited a rapidly growing trend during the unconditioned heating step during the first 30 min, doubling to 0.10 mgGAE/mL with temperature rising from 33 °C (after 5 min of process time) up to 43 °C. In SFN_T2, the growth in TPC was insignificant up to the temperature of 40 °C. This result could be due to slightly higher working temperatures in SFN_T2.

Later on, in SFN_T1, with the further unconditioned heating to 47 °C (45 min of process time), the respective TPC increased abruptly by nearly 100%, up to 0.12 mgGAE/mL, then oscillating between 0.10 and 0.14 mgGAE/mL up to the temperature of 67.5 °C, reached after 90 min of process time. In SFN_T2, during the 30 min of isothermal step at the temperature of 43°C, TPC increased further, up to the level of 0.13 mgGAE/mL, achieved after 60 min of total process time.

The preliminary conclusion can be drawn, that TPC increased with both temperature and cavitation time, but only up to temperature levels of, or below, 47 °C. No peak in TPC was detected for SFN_T1 in Figure 3a, suggesting that a longer cavitation time at the constant temperature of 43 °C could lead to further increase in TPC.

The evolution of the total flavonoids content exhibits very similar, strong and significant growing trends during the unconditioned heating step in the first 30 min, up to the temperature of 40 °C for SFN_T1 (with 0.30 mgCE/mL), and 43 °C for SFN_T2 (with 0.32 mgCE/mL). Later on, in SFN_T2, during the 30 min of isothermal step at the temperature of 43 °C, TFC kept on growing almost linearly in time, up to 0.43 mgCE/mL (60 min of process time), and no evidence of peak arose. In SFN_T1, the TFC growth paused when temperature increased from 40 °C to 47 °C, then grew again, almost linearly in time, up to the level of 0.55 mgCE/mL, achieved at the temperature of 67.5 °C (90 min of total process time), again without an apparent peak.

The preliminary conclusion can be drawn, that TFC definitely increased with both temperature and cavitation time, but the isothermal step at 43 °C seems more effective in the extraction of flavonoids.

### 3.3. Antioxidant Activity

Table 1 and Table 2 shows the antioxidant activity, according to the DPPH and ORAC assays, observed for samples extracted during the tests SFN_T1 and SFN_T2, respectively. Measurements were performed the day after each test.

First, it is remarkable that after only 15 min of process time, both tests resulted in levels of the IC50 for the DPPH antioxidant activity below 20 μg/mL. It is the same level attributed to ascorbic acid [44].

In SFN_T1 (Table 1), the DPPH assay showed a decrease of the respective IC50 level (increase of the antioxidant activity) from the temperature of 40 °C to 47 °C (45 min of process time), down to the lowest level of 10.1 μg/mL. Since then, the DPPH antioxidant activity collapsed exponentially, with the respective IC50 level rising to about 351 μg/mL at the temperature of 67.5 °C (90 min of process time). Based on Figure 3a, the evolution of the DPPH antioxidant activity seemed to reflect the TPC trend up to the temperature of 47 °C, then it decoupled from the latter at higher temperatures. No relationships arose with the flavonoids concentration shown in Figure 3b.

In SFN_T2 (Table 2), the DPPH assay showed a relatively low value of the IC50 level after just 5 min of process time, followed by a moderate decrease (increase of antioxidant activity), but only up to the temperature of 37.5 °C. During the isothermal step at the temperature of 43 °C, a further decrease of the IC50 level occurred during the first 15 min, later stabilizing below 15 μg/mL. Based on Figure 3a,b, the evolution of the DPPH antioxidant activity did not seem to reflect strictly either the TPC or TFC trends, beyond the fact that all three quantities showed a sustained increase (except for DPPH’s IC50 level during the latter part of the process).

As a preliminary conclusion, the DPPH antioxidant activity was extremely sensitive to the temperature level, with a sudden drop beyond 47 °C. It benefitted from both heating and, likely even more, cavitation time at lower temperatures.

Moreover, the concentration of total phenolics showed a moderate correlation with the DPPH antioxidant activity, which is not simple to elucidate. Some authors claimed that phenolic compounds are not the only factors responsible for the antioxidant activity, with other phytochemicals (e.g., carotenoids, terpenes, reducing carbohydrates, and essential oils) potentially affecting the total antioxidant activity. As well, other factors could affect the antioxidant properties of a compound, such as possible synergistic and antagonistic effects among additional components, interactions between the physical environment of the sample and the phenolic compounds, or the activity of specific phenolic compounds that were not suitably determined [42]. Anyway, based on the comparison of the results shown in Figure 3, Table 1 and Table 2, the concentration of total phenolics could positively contribute to the small, yet significant, difference in the highest levels of the DPPH antioxidant activity (lowest IC50), occurring in both tests after 45 min of process time.

The evolution of the ORAC antioxidant activity looks very different from DPPH. In SFN_T1 (Table 1), the ORAC assay showed practically stable levels from 15 to 45 min of process time (temperature increase from 33 to 47 °C). Then, a remarkable growth was observed up to 75 min of process time (temperature of 61.5 °C), followed by a sharp drop in the latter 15 min of process time (up to the temperature of 67.5 °C). Based on Figure 3a,b, the evolution of the ORAC antioxidant activity did not reflect the TPC, nor the TFC trends.

In SFN_T2 (Table 2), the ORAC assay showed stable levels from 5 to 15 min of process time (temperature increase from 33 to 37.5 °C), followed by a sharp increase (more than doubling) in the next 15 min (up to the temperature of 43 °C). Then, in the isothermal step at the temperature of 43 °C, a further remarkable growth of the antioxidant activity occurred, with no apparent peak. At the end of the process, the level of the ORAC antioxidant activity was much greater than the peak level observed in SFN_T1 (about 840 vs. 585 μMTE/L). Based on Figure 3a,b, the evolution of the ORAC antioxidant activity showed correlation with both TPC and TFC, especially for temperatures levels higher than 37 °C, with exponential functions accurately fitting both relationships (R^2^ > 0.92).

The preliminary conclusion can be drawn, that ORAC levels were very sensitive to the temperature, yet differently from DPPH. Indeed, ORAC levels increased with temperature in the approximate range 40 to 60 °C, suddenly dropping with further heating. The ORAC levels also showed a stronger increase with cavitation time at the temperature of 43 °C; the investigation of such dependence on cavitation time at other temperature levels in the aforementioned range should be a subject for further research. Moreover, limited to SFN_T2, the concentration of total phenolics and total flavonoids showed accurate correlations with the ORAC levels, in the temperature range 31.5 to 43 °C, as well as during the isothermal step at the latter temperature level.

### 3.4. Stability

Aimed at assessing the time stability of the silver fir needles aqueous solutions, the TPC, TFC, DPPH, and ORAC levels were measured again, for both tests SFN_T1 and SFN_T2, 47 days and eight days later, respectively. Although other choices would have been possible, the samples collected after 60 min of process time, in each test, were selected for the stability assessment, because they corresponded to the maximum antioxidant activity, both DPPH and ORAC, in SFN_T2; the second peak level of the ORAC, and moderate DPPH in SFN_T1. Moreover, those samples shared the same process time. It is recalled that the operational temperatures after 60 min of process time were 54 °C in SFN_T1, and 43 °C in SFN_T2.

Table 3 shows the antioxidant activities, according to the DPPH and ORAC assays, observed at different times after the tests SFN_T1 and SFN_T2.

With regard to the antioxidant activity, two striking and contrasting features arose. The antioxidant activity measured by the ORAC assay decreased to less than 30% of its level in day 1 in SFN_T1, and to less than 20% in SFN_T2, despite the shorter time lapse between the observations in the latter case. The IC50 dose for the DPPH assay increased by 50% compared to its level in day 1 in SFN_T1 and did not change in SFN_T2.

As mentioned in Section 2.3.3, the DPPH assay is specific for the estimation of the antioxidant activity of plant extracts and is related to activity against lipid oxidation. As mentioned in Section 2.3.4, the ORAC assay is more specific for human plasma, and measures the scavenging of peroxyl as well as of hydroxyl radicals.

Based on the above, the aqueous extracts produced by means of the hydrodynamic cavitation process, absent any preservative additives, fully retained the antioxidant activity towards lipid oxidation, at least up to nine days after processing, and its decay is relatively moderate even after 48 days. Conversely, the ability to scavenge peroxyl and hydroxyl radicals dropped very quickly, though it was still observable after 48 days.

With regard to total phenolics and total flavonoids, the results are relatively uniform across the two tests, showing comparable relative drops in the respective concentrations, once the different time lapses are considered, as well as a continuous decay in time after samples production. Apparently, the HC process did dot damage or denature the compounds responsible for the DPPH antioxidant activity, either polyphenols, flavonoids, or others, such as essential oils, at the condition of limiting the process temperatures, a safe level for the latter being somewhere between 43 and 47 °C.

## 4. Discussion

The observed levels of the in vitro antioxidant activity of the samples extracted during the performed tests highlight the very good potential of the HC process for the creation of aqueous solutions, endowed with bioactive compounds extracted from silver fir needles. In test SFN_T1, the activity towards DPPH radicals, measured the day after the test (Table 1), achieved the lowest level of the respective IC50 (highest antioxidant activity) of about 10 μg/mL, while, in SFN_T2, the lowest level was about 14 μg/mL (Table 2), both after 45 min of process time.

Table 4 shows the highest level obtained for the DPPH antioxidant activity, throughout the tests carried out under this study, compared with results for few reference substances, including a synthetic antioxidant, and vitamin E, catechin, two essential oils, one of which extracted from *Abies alba* twigs and needles, and aqueous extracts from coniferous trees. Despite the relatively wide variability in the published levels of DPPH IC50, even for the same reference substances [45], the respective lowest level of DPPH IC50, achieved in the performed HC-driven tests, competed even with the lower values reported for ascorbic acid, quercetin, and catechin. It was also significantly lower than any other results, including the considered synthetic antioxidant, the essential oils, and other extracts.

The ORAC antioxidant activity levels for the aqueous solutions obtained in this study, from HC processing of *Abies alba* needles, were hard to compare with data obtained from other studies, due to scarce literature. In particular, very few data exist about the ORAC levels of aqueous solutions, such as beverages, while plenty of data are available, referred to the unit mass of several functional substances.

However, at least one quite comprehensive study investigated the ORAC levels of several commercial beverages available on the U.S. market, also in relation with the respective total phenol concentration [49]. That is especially interesting, due to the observed strict relationship between the aforementioned quantities for the SFN_T2 test, where the highest ORAC level was achieved.

Table 5 shows the ORAC and TPC levels for three representative samples extracted from SFN_T2, as well as for most of the commercial beverages investigated in [49]. Despite all the ORAC levels for those beverages were higher than the levels achieved in this study, the ratios between the respective levels for ORAC and TPC were relatively similar.

Figure 4 shows the ORAC to TPC ratios, computed from data shown in Table 5, including the propagation of the respective uncertainties. These data tell an interesting story, with key points summarized as follows:The ORAC to TPC levels found in this study increased with cavitation time, with the SFN_T2 sample collected after 60 min exhibiting both the highest ORAC level and the highest ORAC to TPC level, which could suggest the HC ability to extract more and more functional polyphenols, likely bound in the raw material, during the process.The highest ORAC to TPC level found in this study was comparable with most of the respective levels found for the considered commercial beverages, and it is likely to increase further with longer and/or optimized cavitation process.The ORAC levels achieved in this study are likely to increase also after increasing the concentration of the raw material added to water, which was very low in this study; however, the dose-dependency over the ORAC antioxidant activity needs specific investigation.

Thus, interesting perspectives exist about obtaining aqueous solutions, based on silver fir needles as raw material, and manufactured by means of the HC method proposed in this study, endowed with high ORAC levels. The above considerations also lead to recommending further research on the relationship between the ORAC antioxidant activity and the functional compounds extracted in the aqueous solution.

Based on the nature of the raw material, the structural and operational features of the performed processes, and the obtained results, it is safe to state that the HC technique and the related method, proposed in this study, adhere to the nowadays well-established six principles of green extraction [50]:(1)Innovation by selection of varieties and use of renewable plant resources: *Abies alba* Mill. is a plant species at risk in Italian northern Apennines, relict of past large populations [51]; moreover, fir needles are abundant and renewable by-products of forest management, and can be used in small proportion to achieve remarkable oxidant activity in aqueous solution.(2)Use of alternative solvents and principally water or agro-solvents: water was the only solvent used in the discussed extraction method.(3)Reduce energy consumption by energy recovery and using innovative technologies: as little as 0.04 kWh of electricity per liter of aqueous solution were consumed during 60 min of process time in both tests discussed in this study, with no other energy source used during operation; electricity consumed for centrifuge separation was not accounted for, but it was assumed negligible.(4)Production of co-products instead of waste to include the bio- and agro-refining industry: once deprived of soluble (and solubilized) compounds, the residual fraction of the original mass of fir needle, which had to be separated from the aqueous solution, could be destined to composting, anaerobic digestion, or even to reuse as feedstock for biochar [52].(5)Reduce unit operations and favor safe, robust and controlled processes: the discussed extraction method comprised only two operations after fir needles harvesting, i.e., HC processing, and mechanical separation; the equipment was simple, safe, robust, and easily controllable; the HC process needed to achieve high levels of the antioxidant activity was very fast (60 min or less).(6)Aim for a non-denatured and biodegradable extract without contaminants: absent any additives, water and fir needles were the only ingredients; although indirectly inferred, as discussed in Section 3.4, the HC process did not denature the antioxidant compounds of silver fir needles.

## 5. Conclusions

The results discussed in this study appeared to meet the best expectations raised by other emerging and successful HC applications. Indeed, HC processing of silver fir needles, as single unit operation, was proven to boost the antioxidant activity of the resulting aqueous extract, without the use of any solvent other than water, by means of a very fast process, especially considering the very small concentration of the raw material (0.44% *w*/*w* dry basis).

This study could open a new field of application for controlled HC processes, with potential for practical developments, due to high efficacy, energy and resource use efficiency, as well as straightforward scalability. These properties were the main drivers of several recent and successful implementations of HC processes at the pilot and industrial scale, e.g., in the fields of water heating [53], beer-brewing [20,22], wastewater remediation [27], pretreatment of biomasses [54], and enhancement of biochar properties [36]. The same properties were considered essential for bridging the gap from the pilot to the industrial scale in the food field, such as in the extraction of bioactive compounds [18,55], including from fruits [56], and cereals [57], as well as in food pasteurization and sterilization [18,58], and in the creation of stable oil-in-water nanoemulsions [18].

HC processes could lead to boost the efficacy of *Abies alba* needles in the applications described in Appendix B, as well as to extend the respective use in other fields, such as preserving and health-promoting additives in beer. However, much research remains to be done.

The DPPH antioxidant activity of aqueous solutions was shown to be remarkably dependent on solution pH (higher in less acidic media), and on the nature and concentration of metal ions [39]. A similar dependence can be expected for silver fir needles water extracts, including dependence on the properties of further liquids in which it should be diluted (e.g., in the case of fortification of mineral waters or other beverages, including alcoholic ones). Further research along this direction is therefore recommended. As well, the dose-dependency of the aqueous solution properties should be investigated (only one concentration level was used in this study).

As pointed out in Section 3.4., it is likely that ORAC-related compounds are much more unstable in time than DPPH-related compounds; more generally, the relationship between antioxidant activity, either DPPH or ORAC, and the functional compounds extracted from silver fir needles, needs further investigation. Such recommendation has already been suggested, limited to the ORAC antioxidant activity, in relation to the results shown in Figure 4. Thus, further research is urged, about the accurate, quantitative analysis of the composition of the aqueous solutions, e.g., polyphenols, flavonoids, and essential oils, extracted by means of the method proposed in this study.

Another direction for further research could concern the additivation of silver fir needles water extracts with preservative agents, such as citric acid or related formulations [59], aimed at the stabilization in time of the antioxidant activity. The additivation with pectin from natural sources could also be useful, due to its emulsifying properties, potentially leading to stable encapsulation of essential oils [60]. As well, the investigation of the dependence of the stability on the water properties is recommended.

More generally, only silver fir (*Abies alba* Mill.) needles were processed for the purpose of this study. Whether hydrodynamic cavitation could be successfully applied to other plant raw materials, however likely it may be, should be proven by suitable experiments, thus suggesting another direction for further research.

Last, in this study, both DPPH and ORAC antioxidant activities were measured in vitro. Further research should address the in vivo effectivity of the aqueous solutions of fir needles, produced by means of HC processes, e.g., towards the prevention of lipid oxidation in storage foods, stimulation of plant growth, and fortification of beverages, the latter along with the investigation of the effects on the human health, including the bioavailability issue. In this regard, innovative assays could be considered, to measure the antioxidant activity of food extracts, based on the generation of physiological radical species [61].

## Figures and Tables

**Figure 1 foods-08-00065-f001:**
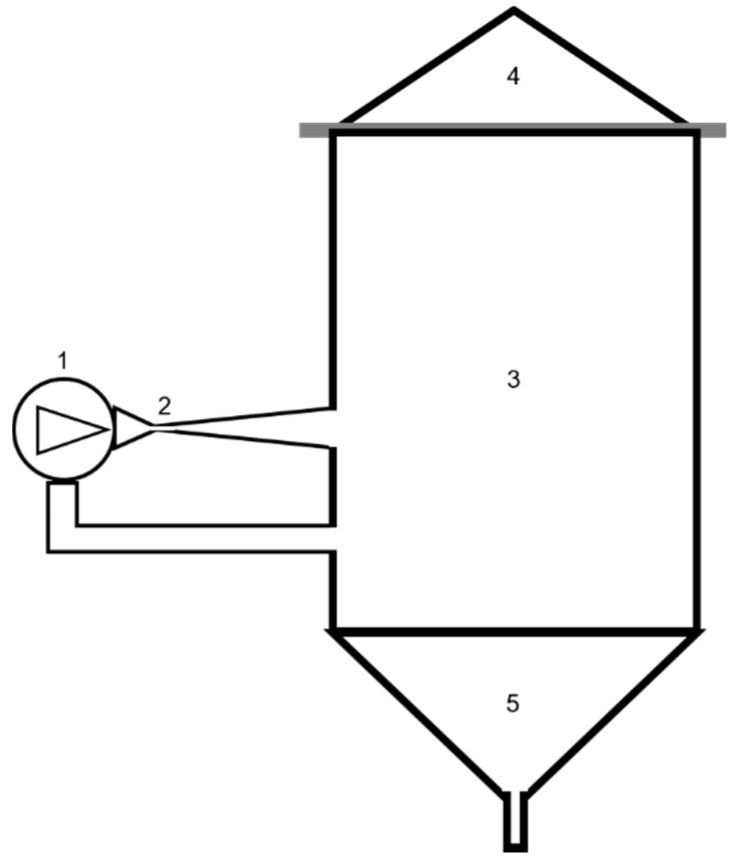
Experimental hydrodynamic cavitation (HC)-based installation. 1—centrifugal pump, 2—HC reactor, 3—main vessel, 4—cover, 5—discharge.

**Figure 2 foods-08-00065-f002:**
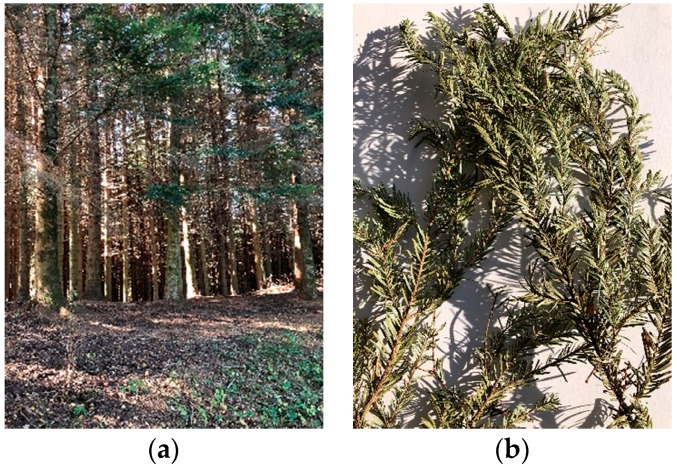
(**a**) View of a portion of the mixed firs forest; (**b**) silver fir twigs and needles.

**Figure 3 foods-08-00065-f003:**
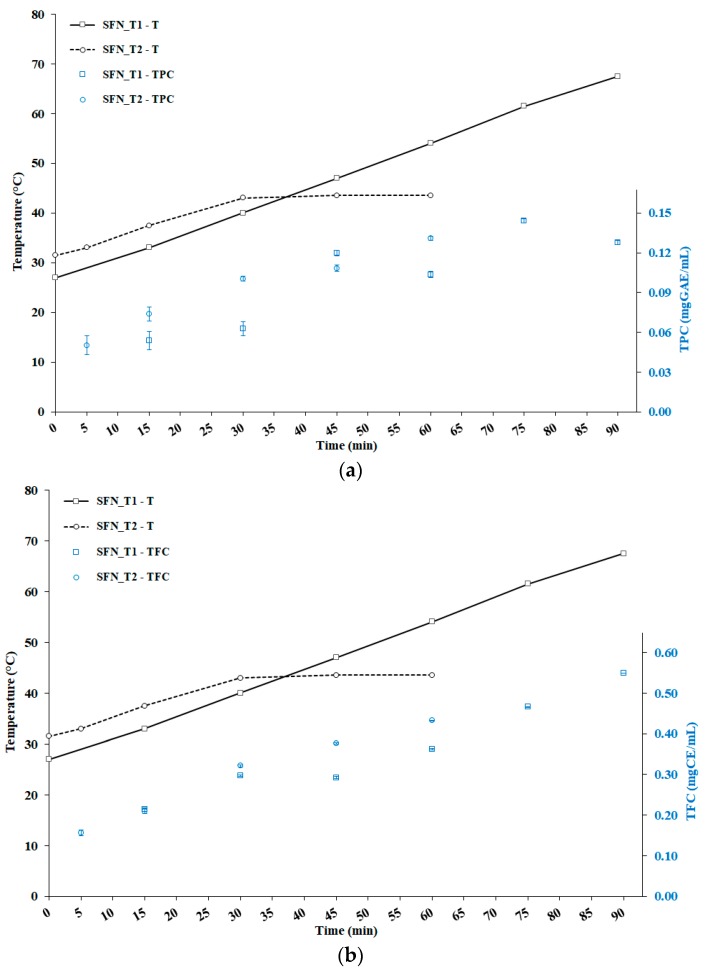
(**a**) Joint evolution of temperature (T) and total phenolic content (TPC) for tests SFN_T1 (Silver Fir Needles—Test 1) and SFN_T2 (Silver Fir Needles—Test 2); (**b**) Joint evolution of temperature (T) and total flavonoids content (TFC) for tests SFN_T1 and SFN_T2. Error bars represent the standard deviations.

**Figure 4 foods-08-00065-f004:**
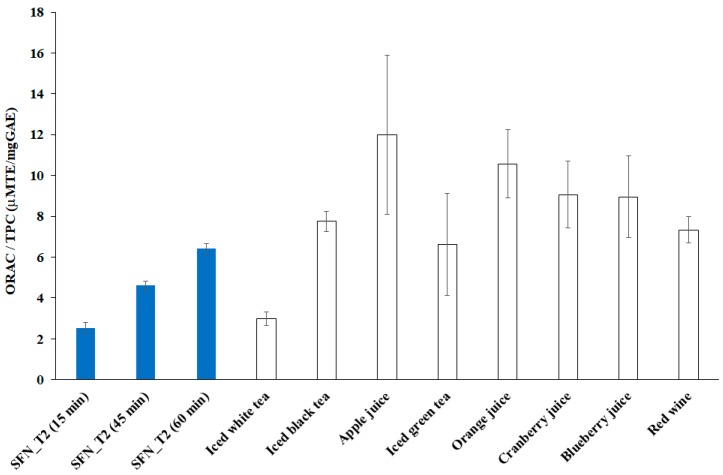
ORAC (The oxygen radical absorbance capacity) to TPC (The total phenolic content) ratios for three samples from the test SFN_T2 (this study) and referred to few commercial beverages.

**Table 1 foods-08-00065-t001:** 2.20-diphenyl-1-picrylhydrazyl (DPPH) and oxygen radical absorbance capacity (ORAC) antioxidant activities, with the respective standard deviations, for test SFN_T1 (Silver Fir Needles—Test 1). Superscripts indicate indistinguishable values (*p* > 0.05).

Time (min)	T (°C)	DPPH (IC50, μg/mL)	ORAC (μMTE/L)
0	27.0		
15	33.0	16.5 ± 1.0 ^a^	201.7 ± 14.0 ^b^
30	40.0	14.4 ± 0.7 ^a^	163.0 ± 11.4 ^b^
45	47.0	10.1 ± 0.4	184.6 ± 12.9 ^b^
60	54.0	44.0 ± 2.1	457.9 ± 24.1
75	61.5	150.5 ± 8.4	585.9 ± 27.1
90	67.5	350.8 ± 23.7	295.4 ± 18.7

IC50: The Effective Concentration values.

**Table 2 foods-08-00065-t002:** DPPH and ORAC antioxidant activities, with the respective standard deviations, for test SFN_T2 (Silver Fir Needles—Test 2). Superscripts, in any column, indicate indistinguishable values (*p* > 0.05).

Time (min)	T (°C)	DPPH (IC50, μg/mL)	ORAC (μMTE/L)
0	31.5		
5	33.0	27.4 ± 1.6	190.6 ± 7.3 ^c^
15	37.5	19.5 ± 0.9 ^a^	186.6 ± 18.1 ^c^
30	43.0	19.5 ± 0.9 ^a^	393.8 ± 25.6
45	43.0	13.7 ± 0.5 ^b^	497.9 ± 22.8
60	43.0	14.7 ± 0.8 ^b^	840.8 ± 31.4

**Table 3 foods-08-00065-t003:** TPC, TFC, DPPH (IC50), and ORAC levels, with the respective standard deviations, measured at different times after tests SFN_T1 and SFN_T2. Data refer to samples collected after 60 min of process time in each test. Superscripts indicate indistinguishable values (*p* > 0.05). Difference expressed as % change compared to the initial value.

	SFN_T1			SFN_T2		
	Day 1	Day 48	Diff.	Day 1	Day 9	Diff.
**TPC** (mgGAE/mL)	0.103 ± 0.002	0.053 ± 0.007	−48%	0.131 ± 0.002	0.091 ± 0.003	−31%
**TFC** (mgCE/mL)	0.363 ± 0.002	0.209 ± 0.004	−42%	0.432 ± 0.001	0.309 ± 0.002	−28%
**DPPH** (IC50, μg/mL)	44.0 ± 2.1	65.8 ± 3.0	50%	14.7 ± 0.8 ^a^	14.4 ± 1.0 ^a^	0%
**ORAC** (μMTE/L)	457.9 ± 24.1	128.3 ± 8.5	−72%	840.8 ± 31.4	152.3 ± 5.7	−82%

The total phenolic content (TPC); The total flavonoids content (TFC); Diff.: Difference.

**Table 4 foods-08-00065-t004:** DPPH antioxidant activity: the highest level (lowest IC50) found in this study, and levels found for other substances. Where available, the respective standard deviations are indicated.

Substance	DPPH (IC50, μg/mL)	Ref.
*Abies alba* needles extract	10.1 ± 0.4	This study ^a^
Ascorbic acid (reference substance)	5.85	[45]
Ascorbic acid (reference substance)	7.62	[12]
Ascorbic acid (reference substance)	20 ± 1.3	[44]
Ascorbic acid (reference substance)	50	[45]
Resveratrol (reference substance)	16.62	[12]
Quercetin (reference substance)	10.5 ± 4.6	[46]
Butylated hydroxytoluene (synthetic antioxidant, reference substance)	11.58	[12]
Butylated hydroxytoluene (synthetic antioxidant, reference substance)	21.30	[45]
α-Tocopherol (vitamin E)	27.1	[45]
Epigalocatechin gallate (a type of catechin)	7.06	[12]
*Abies alba* twigs and needles (essential oil)	27 ± 6.3	[44]
Clove (essential oil)	13.2 ± 2.9	[46]
*Abies alba* wood (extract)	35.46	[12]
*Pinus coulteri* needles (extract) ^b^	22.7 ± 0.6	[47]
*Pinus densiflora* needles (extract) ^c^	270	[48]

^a^ Lowest level of DPPH IC50 observed throughout the tests. ^b^ Crude extract. ^c^ Hot water extract.

**Table 5 foods-08-00065-t005:** ORAC antioxidant activity and TPC levels, with the respective standard deviations, from this study, and from a previous study (the TPC levels were referred to the volume of 1 L).

Substance	ORAC (μMTE/L)	TPC (mgGAE/L)	Ref.
*Abies alba* needles extract	186.6 ± 18.1	74 ± 4	This study ^a^
*Abies alba* needles extract	497.9 ± 22.8	108 ± 2	This study ^b^
*Abies alba* needles extract	840.8 ± 31.4	131 ± 2	This study ^c^
Iced white tea	2700 ± 300	900 ± 0	[49]
Iced black tea	3100 ± 200	400 ± 0	
Apple juice	4800 ± 1000	400 ± 100	
Iced green tea	5300 ± 1900	800 ± 100	
Orange juice	7400 ± 500	700 ± 100	
Cranberry juice	15,400 ± 2100	1700 ± 200	
Blueberry juice	20,600 ± 2900	2300 ± 400	
Red wine	25,700 ± 2100	3500 ± 100	

^a^ 15 min of process time. ^b^ 45 min of process time. ^c^ 60 min of process time.

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
