# Peer review of "Affordable Production of Antioxidant Aqueous Solutions by Hydrodynamic Cavitation Processing of Silver Fir (Abies alba Mill.) Needles"

_foods, 2019, doi:10.3390/foods8020065_

Round 1
Reviewer 1 Report
The authors have satisfactorily responded to all my questions and made the necessary changes to the manuscript.
Author Response
Response to Reviewer 1 Comments
Point 1: The authors have satisfactorily responded to all my questions and made the necessary changes to the manuscript.
Response 1: The authors highly appreciate this comment by the esteemed Reviewer, and are glad to have successfully accomplished their tasks.
Reviewer 2 Report
The title of the publication is unambiguous and concerns one vegetable raw material. The presented research hypotheses are reasonably justified. In my opinion, the introduction is interesting but too long. Attention is also given to the discussion, too far from the main topic. It should be significantly shortened, especially since I have doubts about the purpose of presenting not only my own research hypotheses.
Author Response
Response to Reviewer 2 Comments
Point 1: The title of the publication is unambiguous and concerns one vegetable raw material. The presented research hypotheses are reasonably justified.
Response 1: The authors thank very much the esteemed Reviewer for these comments, as well as are very glad for her/his appreciation.
Point 2: In my opinion, the introduction is interesting but too long.
Response 2: The authors thank the esteemed Reviewer for this comment. Indeed, the Introduction section was a bit too long. It has been shortened and streamlined in the revised version (737 words, against 848 words in the original version).
Point 3: Attention is also given to the discussion, too far from the main topic. It should be significantly shortened, especially since I have doubts about the purpose of presenting not only my own research hypotheses.
Response 3: The authors thank very much the esteemed Reviewer for this comment. Indeed, the Discussion section was far too long and, especially, too far from the main topic. In the revised version, it has been drastically shortened and focused on the main topic (1,309 words, against 2,194 words in the original version, including Tables and Figure captions), as well as a very small part of the original text has been moved to the Conclusions section.
Further note by the authors: A couple of amendments have been made to the text: in the Conclusions section (adding a short statement and related Reference, concerning additivation for product stability), and in Appendix A (deleting a useless statement). As well, very few words were replaced by more appropriate ones throughout the text, without affecting the content in any way.
This manuscript is a resubmission of an earlier submission. The following is a list of the peer review reports and author responses from that submission.
Round 1
Reviewer 1 Report
This is overall a complete manuscript, with good writing. Just two minor suggestions:
The introduction is too tedious. Please simply it.
For section 2.4.1, please provide more information including such as how much sample was used for the reaction, the reaction time, and the standard curve range.
Reviewer 2 Report
There are many areas to be improved.
Introduction should be written in succinct and relevant.
Materials and Methods:
1. No statistical methods described.
2. Section 2.2 is not a method and should not be there
3. Are you sure that the sample SFN_T1 and SFN_T2 are representative? How about the experimental procedures you described to obtain your results?
4. Are you sure all the absorbance studies carried out were exclusive for each of the group of compounds you intend to assay? For e.g. total phenolic content and total flavonoid content. Some flavonoids can be phenolic compounds also.
Discussion
Due to the above concerns, Discussion is not supported. Hence, any conclusion is hard to be drawn.